# Comparative Transcriptome Analysis Reveals Mechanisms of Differential Salinity Tolerance Between *Suaeda glauca* and *Suaeda salsa*

**DOI:** 10.3390/genes15121628

**Published:** 2024-12-19

**Authors:** Qidong Yan, Shang Gao, Xianglun Zhang, Guoping Liu, Peitao Chen, Xuanyi Gao, Li Yuan, Yucheng Tian, Dapeng Li, Xuepeng Zhang, Huan Zhang

**Affiliations:** 1Shandong Academy of Agricultural Sciences, Jinan 250100, China; xianglunzhang@163.com (X.Z.); xuanyigao2010@126.com (X.G.); yuanlif@163.com (L.Y.); lidp1990@126.com (D.L.); 13001297977@163.com (X.Z.); 2Key Laboratory of Sweet Potato Biology and Biotechnology, Ministry of Agriculture and Rural Affairs/Beijing Key Laboratory of Crop Genetic Improvement/Laboratory of Crop Heterosis & Utilization and Joint Laboratory for International Cooperation in Crop Molecular Breeding, Ministry of Education, College of Agronomy & Biotechnology, China Agricultural University, Beijing 100193, China; 18708615431@163.com (Q.Y.); capital0529@163.com (P.C.); 3Shandong Bohua High-Efficient Ecological Agriculture Science & Technology Co., Ltd., Binzhou 256506, China; m17863805315@163.com; 4Key Laboratory of Livestock and Poultry Multi-Omics of MARA, Institute of Animal Science and Veterinary Medicine, Shandong Academy of Agricultural Sciences, Jinan 250100, China; 5Kenli Bureau of Agriculture and Rural Affairs, Dongying 257599, China; dyklzcfg2022@163.com

**Keywords:** *Suaeda glauca*, *Suaeda salsa*, saline–alkali stress, comparative transcriptome

## Abstract

Background: *Suaeda glauca* and *Suaeda salsa* have obvious morphological features and strongly tolerate saline–alkali environments. However, the mechanisms that lead to the differences in saline–alkali tolerance between them remain unclear. Methods: In this study, we employed comparative transcriptome analysis to investigate *S. glauca* and *S. salsa* under saline–alkali stress. Results: Our sequencing efforts resulted in the identification of 99,868 unigenes. We obtained 12,021 and 6227 differentially expressed genes (DEGs) from the *S. glauca* and *S. salsa* under salt stress compared with plants in the control. Notably, 1189 and 1864 were specifically upregulated DEGs in the roots and leaves of *S. salsa* under saline–alkali conditions, respectively. These genes were enriched in pathways such as “Plant hormone signal transduction”, “Carbon metabolism” and “Starch and sucrose metabolism”. Further analysis of stress-related pathways and gene expression levels revealed that key genes involved in abscisic acid (ABA) and jasmonic acid (JA) biosynthesis, ABA signal transduction, and their downstream transcription factors were upregulated in the roots of *S. salsa* under saline–alkali conditions. Additionally, 24 DEGs associated with stress response were identified in the roots and leaves of both species. The expression levels of these pathways and related genes were higher in *S. salsa* than in *S. glauca*, suggesting that *S. salsa* enhances its saline–alkali tolerance by elevating the expression of these genes. Conclusions: This study provides a new research perspective for revealing the differences in saline–alkali tolerance mechanisms between *S. glauca* and *S. salsa*, bringing forth important candidate genes for studying their saline–alkali tolerance.

## 1. Introduction

Soil salinization has become a global problem, affecting approximately 7% of the land area worldwide [1]. More than 20% of 230 million hectares of globally irrigated land suffer from some degree of soil salinization. This proportion continues to grow due to climate impact and unreasonable agricultural irrigation, posing a huge threat to the sustainability of crop production [2]. Soil salinization is caused by multiple factors determined by natural processes and human activities [3]. First, the primary salinization caused by natural factors, such as temperature rise, sea level rise, and saltwater intrusion into freshwater aquifers, has adverse effects on farmlands in some coastal cities [4]. Second, the freshwater shortage of irrigation and the continuous degradation of agricultural land due to salt stress have led to significant losses in agricultural productivity [5]. The problem of soil salinization is becoming increasingly serious in China. China has about 93 million hectares of salinized soil, accounting for 10% of the total arable land area, severely affecting crop production and food security [6].

When the soil’s nutrients and water potential are altered, harmful ions can accumulate under saline–alkali stress. These factors can cause ion toxicity and osmotic, oxidative, and pH stresses on plants, affecting plant growth and even leading to severe plant death [7]. High salinity can decrease soil water potential, resulting in osmotic stress on plants, causing water leakage, and reducing their absorption capacity of water and mineral elements, such as potassium and calcium ions [8]. Plants can reduce the harm of osmotic stress by accumulating osmoregulatory substances (e.g., proline, glycine, betaine, and sugar) [9]. The continuous accumulation of salt ions in soil can cause plant toxicity. For example, the accumulation of sodium ions can disrupt the balance of sodium and potassium ions in cells. Sodium ions can also harm metabolic activities in the cytoplasm and cell membranes, leading to ion toxicity [10].

Under high pH stress, the structure and function of plant roots can be seriously affected. The permeability of plant cell membranes can also be influenced in these conditions, leading to the influence of the absorption and metabolism of calcium ions, magnesium ions, and other elements in plants, breaking the ion balance inside cells. Plants under ion stress maintain ion homeostasis by transporting them inside and outside their cells, thus reducing the impact of ion toxicity on themselves [11]. Plants can achieve ion homeostasis through various pathways, such as the salt hypersensitive (SOS) signaling pathway in *Arabidopsis*, which has been shown to export Na^+^ outward. The key genes in this pathway are *SOS1*, *SOS2*, and *SOS3* [12]. High-affinity potassium transporters (HKTs) also play an important role in plant salt tolerance as they can transport sodium ions from plant stems to other tissues, enhancing plant saline–alkali tolerance [13]. Proton pumps [14] and Na^+^/H^+^ antiporters (NHX) [15] also play important roles in maintaining ion homeostasis in plant cells. The accumulation of reactive oxygen species (ROS) under saline–alkali stress can cause irreversible oxidative damage to plant membranes, proteins, nucleic acids, and other substances, affecting many plant metabolic functions. Antioxidant plant enzymes, e.g., catalase (CAT), peroxidase (POX), superoxide dismutase (SOD), and ascorbate peroxidase (APX)] and nonenzymatic antioxidant molecules, e.g., glutathione (GSH), carotenoids, and ascorbic acid (ASC)] may remove accumulated ROS in plants at certain levels, reducing oxidative stress damage [16]. Despite several negative effects caused by saline–alkali stress on plants, they can alleviate them through various regulatory actions to ensure their growth and development.

Although many plants cannot grow normally under saline–alkali stress conditions, some species still have a natural saline–alkali tolerance, growing normally in high saline–alkali environments, such as alkali grass [17], wild soybeans [18], white thorns [19], and other alkali-tolerant species [20]. These species are key materials for studying the mechanism of plant saline–alkali tolerance. *S. glauca* and *S. salsa*, annual halophilic herbaceous plants belonging to the *Chenopodiaceae* family, are typical halophilic plants distributed in Europe and Asia. These species with strong saline–alkali tolerance are widely produced in Northeast, North, and Northwest China in provinces like Shandong, Jiangsu, and Zhejiang [21,22]. As model halophytes containing multiple nutrients, these plants have received widespread attention in livestock raw material production, food processing, industrial production, and soil improvement [23,24,25]. Although both *S. glauca* and *S. salsa* can grow in saline–alkali soils, they still differ in their salt tolerance. Since the 1990s, Chinese scholars have conducted a series of studies on the salt adaptability of *Suaeda* plants. However, most of the work has been conducted on *S. salsa*. The research mainly focuses on the physiological response, saline–alkali tolerance mechanism analysis, and molecular levels, such as leaf photosynthetic oxygen release rate, antioxidant enzyme activity, determination of osmotic regulatory substances such as betaine and proline, and screening of genes related to saline–alkali tolerance [26,27,28]. Although *S. glauca* and *S. salsa* are annual salt-tolerant halophilic, their habitats and saline–alkali tolerance differ. Currently, there are relatively few comparative studies on the saline–alkali tolerance mechanisms of *S. glauca* and *S. salsa*. Under saline–alkali stress conditions, both *S. glauca* and *S. salsa* can alleviate the impact of salt stress through means such as leaf succulence, ion compartmentalization, osmotic adjustment, and the antioxidant system [29,30,31]. Salt treatment can significantly promote the absorption of Na^+^ in the aboveground parts of *S. salsa* and *S. glauca*, and significantly reduce the absorption of K^+^ in the aboveground parts. Moreover, the content of Na^+^ in the aboveground parts of *S. salsa* is significantly higher than that of *S. glauca*, while the content of K^+^ is significantly lower than that of *S. glauca*, which demonstrates the differences in their salt tolerance mechanisms [32]. Different salt concentrations have different effects on the growth and leaf succulence of *S. salsa* seedlings, among which Na^+^ and Cl^−^ are the two main factors promoting the succulence of *S. salsa* leaves [33]. Compared with *S. glauca*, *S. salsa* accumulates more total nitrogen (TN), total organic nitrogen carbon (TOC), sodium (Na), and other materials in its leaves. During the process of adapting to salinity, the central metabolism of *S. glauca* is nitrogen metabolism, while that of *S. salsa* is characterized by organic acid metabolism [34]. During the growth period of *S. salsa*, a large amount of Na^+^ will accumulate in its leaves. *S. salsa* can regulate the Na^+^ balance in its body and maintain the accumulation level of Na^+^ through *SsSOS1* and *SsNHX1* [35]. The localization analysis of Na^+^ in *S. salsa* after salt stress revealed that it isolates most of the Na^+^ absorbed from the environment in the vacuoles to maintain the normal ionic balance within cells [36]. Under high Na^+^ or high K^+^ conditions, *S. salsa* has a specific absorption system for K^+^. While absorbing a large amount of Na^+^, it can also effectively absorb K^+^. This characteristic of K^+^ absorption by *S. salsa* is one of the important factors for its normal growth under high salinity conditions [37].

As an important branch of systems biology, transcriptomics has experienced rapid development in recent years, advancing the understanding of gene expression regulatory networks. In this field, high-throughput sequencing technology is used to analyze all transcripts of a certain organism or cell under specific conditions, thereby revealing dynamic changes in gene expression [38]. From early microarray technology to widely used second-generation sequencing (RNA-seq) and emerging third-generation sequencing technologies, transcriptomics has continuously innovated its research tools [39]. RNA-seq has become the mainstream method for transcriptome analysis due to its high sensitivity, high resolution, and lack of need for predesigned probes. RNA-seq technology has been widely applied in various life science fields and has made tremendous progress in basic, medical, and agricultural research [40]. RNA-seq technology plays an important role in studying plant salt tolerance mechanisms. RNA-seq has been widely applied in the research of salt tolerance mechanisms in plants such as rice [41], corn [42], cotton [43], eggplant [44], and sweet potato [45].

*S. glauca* and *S. salsa* are pioneer plants that repair saline–alkali soil [46]. Most research studies involving these two species have focused on their nutritional value, technological applications, salt resistance, and ecological significance instead of transcriptomic perspectives. However, few studies have evaluated their differences in saline–alkali tolerance mechanisms. Therefore, this study analyzed the response of *S. glauca* and *S. salsa* to saline–alkali stress at the transcriptomic level using RNA-seq technology. The metabolic pathways and key genes involved in the response of *S. glauca* and *S. salsa* to salt stress were comprehensively analyzed, providing theoretical support for revealing their resistance mechanisms against saline–alkali stress.

## 2. Materials and Methods

### 2.1. Plant Materials

The materials used in this experiment were *S. glauca* and *S. salsa*, grown in normal soil (control group, CK) and saline–alkali soil (treatment group, T) approximately 2 km south of Liujia Village, Dongji Town, Kenli District, Dongying City, Shandong Province (118°23′20″ E, 37°28′4″ N). *S. glauca* and *S. salsa* from two different environments were collected during the same period. After morphological observation, their leaf (L) and root (R) tissues were separated, washed with pure water, dried, and then quickly frozen with liquid nitrogen and stored at −80 °C in an ultra-low temperature freezer for RNA extraction. This experiment employed three biological replicates (Sg-CK-L1 Sg-CK-L2, Sg-CK-L3, Sg-CK-R1, Sg-CK-R2, Sg-CK-R3, Sg-T-L1, Sg-T-L2, Sg-T-L3, Sg-T-R1, Sg-T-R2, Sg-T-R3, Ss-CK-L1, Ss-CK-L2, Ss-CK-L3, Ss-CK-R1, Ss-CK-R2, Ss-CK-R3, Ss-T-L1, Ss-T-L2, Ss-T-L3, Ss-T-R1, Ss-T-R2, and Ss-T-R3). Each sample showed a normal growth state in the sampling environment (the individuals of *S. glauca* and *S. salsa* in saline–alkali environments were smaller than those in normal environments).

### 2.2. Extraction and Quality Testing of RNA

The leaves and roots of *S. glauca* and *S. salsa* grown in normal and saline–alkali soils were selected, with three biological replicates for each species. TIANGEN’s RNAprep pure total RNA extraction kits were used to extract total RNA (Beijing, China). All vessels used in the experiment were subjected to RNA enzyme inactivation treatment. After total RNA extraction, the concentration of RNA was detected using NanoDrop (Thermo Fisher Scientific, Santa Clara, CA, USA). The integrity and purity of RNA were accurately evaluated using Qubit2.0 and Agilent 2100.

### 2.3. Construction of RNA-Seq Sequencing Library and Transcriptome Sequencing

After sample inspection, library construction and quality control were performed by the BioMarker Technologies Company (Beijing, China). The cDNA library was subjected to transcriptome sequencing using the Illumina NovaSeq 6000 sequencing platform (San Diego, CA, USA). The sequencing read length was PE150.

### 2.4. Transcriptome Assembly and Functional Annotation

The raw sequence obtained from the sequencing was filtered to obtain clean reads. The filtering conditions were as follows: (1) reads with adapters were removed; (2) reads with a proportion of N greater than 0.1% (N indicates the inability to determine base information) were discarded; (3) low-quality reads (reads with quality values Qphred < 20 accounting for more than 50% of the total reads) were filtered out; (4) low-quality reads (reads with quality values Qphred < 20 accounting for more than 50% of the total reads) were removed. Clean reads were obtained for subsequent analysis through the above steps. Trinity (v2.5.1) was used to assemble clean reads of the leaves and roots of *S. glauca* and *S. salsa* grown in two environments to construct the unigene library.

Blast comparison analysis and unigene functional annotation were conducted through public databases KOG/COG (euKaryotic ortholog groups/clusters of orthologous groups of proteins), KEGG, GO, Pfam (protein family), Swiss-Prot, TrEMBL, eggNOG, and NR (NCBI non-redundant protein sequences).

### 2.5. Differential Expression Gene Analysis

Fragments per kilobase of transcript per million mapped reads (FPKM) was applied to estimate gene expression levels. FPKM can eliminate the influence of gene length and sequencing quantity differences on calculating gene expression. It represents the number of reads per kilobase length from alignment to a certain gene per million reads. We used DESeq2 (1.30.1) software to screen DEGs (screening criteria *p*-value < 0.01 and fold change (FC) ≥ 1.5). A Venn diagram of DEGs was drawn using an online platform for data analysis and visualization (https://www.bioinformatics.com.cn; accessed on 23 July 2024). GO enrichment analysis and KEGG pathway enrichment analysis (with a significant enrichment criterion of corrected *p*-value < 0.05) were conducted using topGOR (2.48.0) and KOBAS (v3.0).

### 2.6. Statistical Analysis

The mean and standard error of biological replicates were calculated using Microsoft Excel 2019. Variance (ANOVA) and correlation were conducted using SPSS software (IBM SPSS Statistics version 19, Chicago, IL, USA)

## 3. Results

### 3.1. Morphological Analysis of S. glauca and S. salsa

Both *S. glauca* and *S. salsa* exhibit robust saline–alkali tolerance, enabling them to thrive in soils with such features as seashores, saline–alkali lakes, and salt marshes (Figure 1A). They may dominate single-species plant communities in some instances, while in others, they coexist with other salt-tolerant species to form intricate plant communities (Figure 1B).

*S. glauca* and *S. salsa* display marked similarities in their morphological characteristics and growth habits. Both species possess elongated, fleshy leaves arranged alternately on slender stems without petioles, and their stems are cylindrical, with distinct ridges. They exhibit abundant branching fibrous roots without prominent taproots and bear both hermaphroditic and female flowers. Upon maturity, their seeds are biconvex, black, and lustrous. Both plants thrive in high-humidity environments, demonstrating remarkable tolerance to salt and alkali, adaptability to barren soils, and resistance to pests and diseases. Furthermore, they exhibit a wide range of temperature tolerance. The above-ground parts of both species can be used for animal feed, food, and medicine (Figure 2A,B). Still, both species also have unique traits. For instance, the cross-section of the leaves of *S. salsa* is semicircular or crescent-shaped, while the cross-section of the leaves of *S. glauca* is more circular. Additionally, the branching ability of *S. salsa* is stronger than that of *S. glauca*, while its upright ability is weaker than that of *S. glauca*.

*S. salsa* demonstrates greater adaptability to high-salinity environments than *S. glauca* (the intertidal distribution at the river outlet is exclusively *S. salsa*). Additionally, *S. salsa* exhibits greater waterlogging tolerance and can withstand longer periods of flooding (although it will also present a severe purple-red stress state, Figure 1C) compared to *S. glauca*. However, *S. salsa* has weaker competitiveness in plant communities with lower salinity, and its distribution is significantly higher than that of *S. glauca* in more abundant plant communities.

### 3.2. Quantification of Sequencing Libraries and Transcriptome Assembly

This study used Illumina RNA-Seq technology to sequence the root and leaf samples of *S. glauca* and *S. salsa* grown in normal and saline–alkali soil. This effort successfully obtained expression profile sequence information of the root and leaf tissues of both species under different growth conditions (Appendix A). The raw data obtained from sequencing underwent base calling to convert it into sequence data (raw reads). The depth and coverage of the sequenced reads were calculated to assess the quality of the sequences. The sequencing error rate at individual base positions was lower than 1%, the GC content was approximately 45%, and no AT or GC separation phenomena were observed (Appendix A). After quality control of sequencing, a total of 155.84 Gb of clean data were obtained, with a Q30 base percentage of no less than 93.61% across all samples. This indicated a high base recognition accuracy and adequate sequencing result quality. Thus, the transcriptome sequencing results met the quality requirements for subsequent assembly.

The unigene library assembled from 24 samples contained 99,868 unigenes with an average length of 993.35 bp. The N50 of the unigenes was 2205, indicating high-assembly completeness. Among these, unigenes longer than 500 bp accounted for 13.29% of the total in the library. Unigenes longer than 1000 bp comprised 12.34% of the total, while those exceeding 2000 bp comprised 16.29% (Table 1).

### 3.3. Functional Annotation of Unigenes

The unigene sequences were aligned with the corresponding protein or nucleotide sequences housed in various databases after using BLAST, including KOG, COG, Kyoto Encyclopedia of Genes and Genomes (KEGG), gene ontology (GO), Pfam, Swiss-Prot, eggNOG, TrEMBL, and NR. As a result, 52,527 unigenes were successfully annotated, comprising 52.60% of the overall unigene pool. The annotation statistics are presented in Table 2.

### 3.4. Statistics of Unigene Gene Expression

The correlation analysis of biological replicates is a pivotal indicator for validating the reliability of experiments and samples. We established three biological replicates, and our comparative examination of the correlations in expression levels across all samples revealed strong correlations between the duplicate samples of each treatment. Notably, the Pearson correlation coefficient (r^2^) surpassed 0.85 between repeats of distinct samples, attesting to the robust reproducibility among the three biological replicates (Appendix A). Box plots demonstrate the dispersion of gene expression levels across individual samples and facilitate a visual comparison of the overall gene expression abundance between different samples. A statistical comparison of gene expression levels across samples of *S. glauca* and *S. salsa* under varying growth conditions revealed that, under normal conditions, *S. salsa* exhibited higher gene expression levels in both roots and leaves than *S. glauca*. Notably, *S. glauca* and *S. salsa* displayed elevated overall gene expression in their roots and leaves under saline–alkali conditions, surpassing their respective control samples (Appendix A). This observation underscores the variation in expression levels among samples of *S. glauca* and *S. salsa* from the perspective of the overall dispersion of expression quantities.

### 3.5. Screening of Differentially Expressed Genes (DEGs)

We conducted differential expression analysis across sample groups and compiled statistics on the DEGs in the roots and leaves of *S. glauca* and *S. salsa* plants under diverse growth conditions using DESeq2. *S. glauca* displayed 9114 DEGs in the roots (Figure 3A) and 2907 DEGs in the leaves (Figure 3C) under saline–alkali conditions compared to their control groups. Analogously, *S. salsa* revealed 2808 DEGs in the roots (Figure 3B) and 3419 DEGs in the leaves (Figure 3D) when subjected to saline–alkali conditions compared to its control. A comprehensive list of the DEGs identified in both the roots and leaves of *S. glauca* and *S. salsa* is furnished in Appendix A.

### 3.6. DEG Analysis

*S. glauca* and *S. salsa* exhibited 4758 and 1317 upregulated DEGs in their roots, respectively, compared to the control group. Notably, *S. salsa* roots contained 1189 specifically upregulated DEGs (Figure 4A). Similarly, there were 1185 and 1888 upregulated DEGs compared to the control, with *S. salsa* leaves harboring 1864 specifically upregulated DEGs (Figure 4B).

The 1189 specifically upregulated DEGs in the roots and 1864 in the leaves of *S. salsa* were analyzed using GO and the KEGG pathways. The GO analysis classified the genes into three major categories: “biological process”, “molecular function”, and “cellular component”. Regarding the GO analysis of upregulated DEGs in *S. salsa* roots, the three high-ranking biological processes within the “biological process” category were “metabolic process” (390 DEGs), “cellular process” (356 DEGs), and “single-organism process” (253 DEGs). Within the “cellular component” category, the DEGs were highly abundant in “cell” (324 DEGs), “cell part” (324 DEGs), and “membrane” (322 DEGs) pathways. The three high-ranking molecular functions were “binding” (448 DEGs), “catalytic activity” (374 DEGs), and “transporter activity” (85 DEGs) (Figure 4C). The KEGG pathway enrichment analysis classified the specifically upregulated DEGs in roots into 108 KEGG pathways. Under saline–alkaline conditions, *S. salsa* roots exhibited significant enrichment of specifically upregulated DEGs in several pathways (Figure 4D), including “ribosome” (47 DEGs), “starch and sucrose metabolism” (20 DEGs), “carbon metabolism” (19 DEGs), “plant hormone signal transduction” (18 DEGs), “glycerophospholipid metabolism” (16 DEGs), “biosynthesis of amino acids” (13 DEGs), and “oxidative phosphorylation” (12 DEGs).

In the GO analysis of upregulated DEGs in *S. salsa* leaves, the top three biological processes within the “biological process” category were “metabolic process” (590 DEGs), “cellular process” (560 DEGs), and “single-organism process” (408 DEGs). Within the “cellular component” category, the DEGs were highly abundant in “cell” (359 DEGs), “cell part” (359 DEGs), and “membrane” (474 DEGs) pathways. The top three molecular functions were “binding” (729 DEGs), “catalytic activity” (683 DEGs), and “transporter activity” (77 DEGs) (Figure 4E). In the KEGG pathway enrichment analysis, the specifically upregulated DEGs in the leaves were classified into 124 KEGG pathways. Under saline–alkaline conditions, *S. salsa* leaves exhibited significant enrichment of specifically upregulated DEGs in several pathways (Figure 4F), including “plant hormone signal transduction” (51 DEGs), “phenylpropanoid biosynthesis” (38 DEGs), “starch and sucrose metabolism” (36 DEGs), “biosynthesis of amino acids” (30 DEGs), “Amino sugar and nucleotide sugar signaling” (26 DEGs), “Carbon metabolism” (24 DEGs), and “MAPK signaling pathway-plant” (24 DEGs).

### 3.7. The Important Role of Plant Hormone-Related Genes in Saline–Alkali Stress

The significant role of plant hormones in saline–alkali stress has been widely reported. The KEGG enrichment analysis results indicate that DEGs are notably enriched in the “plant hormone signal transduction” pathway (Figure 4). These results suggest that plant hormones play a crucial role in the response of *S. glauca* and *S. salsa*. to salt and alkali stress. We analyzed the genes related to abscisic acid (ABA) and jasmonic acid (JA) biosynthesis, and ABA signal transduction in their roots to investigate the defensive mechanisms of plant hormones against salt and alkali stress in *S. salsa* and *S. glauca*. We mapped the ABA biosynthesis pathway, ABA signal transduction pathway (Figure 5), and JA biosynthesis pathway (Figure 6) based on the expression patterns of these genes in *S. glauca* and *S. salsa*. Detailed information on the expression levels of genes related to these pathways is provided in Appendix A.

The four upregulated genes identified in the ABA biosynthesis pathway are *ABA4*, *NCED*, *ABA2*, and *AAO3*. *S. salsa* exhibited significantly higher expression levels of these four genes than *S. glauca* when analyzing the expression levels of these genes in the roots of *S. glauca* and *S. salsa*. This result suggests that *S. salsa* might enhance its saline–alkali tolerance by upregulating the genes involved in ABA biosynthesis (Figure 5A). Next, analysis was conducted on genes related to ABA signal transduction, and six upregulated genes were identified. Among the identified genes, one was related to *PYR*/*PYL*, three to the ABA coreceptor *PP2C*, and two to the threonine/serine receptor kinases *SnRK2* and *ABF*. Furthermore, eleven DEGs were identified as functioning transcription factors in response to saline–alkali stress, including three related to *ERF*, three to WRKY, and five to *bHLH* (Figure 5B). The expression levels of these six genes and eleven transcription factors in *S. salsa* were higher than in *S. glauca* (Figure 5B).

An analysis was conducted on genes related to JA biosynthesis. Seven genes with upregulated expression were identified. These genes include one gene associated with *LOX*, two to *AOS*, one to *AOC*, one to *OPR3*, one to *OPR2*, and one to *JAR1*. The expression levels of these seven genes were analyzed in *S. glauca* and *S. salsa*. The expression levels of these genes in *S. salsa* were higher than that in *S. glauca*. This suggests that *S. salsa* might enhance salt and alkali tolerance by upregulating JA biosynthesis-related genes (Figure 6).

### 3.8. Identification of Candidate Genes for Saline–Alkali Tolerance

The upregulated DEGs in the roots and leaves of *S. glauca* and *S. salsa* under saline–alkali stress were screened and identified to further elucidate the differences in their mechanisms of saline–alkali tolerance between *S. glauca* and *S. salsa*. A sum of 17 differentially expressed genes (DEGs) were successfully identified in the roots of *S. glauca* and *S. salsa*, including genes such as *GPAT*, *GDSL Esterases*/*Lipases*, *KCS11*, *TSJT1*, *Cytochrome P450*, *SBT5.6*, *SKOR*, and *SWEET14* (Figure 7A). In the leaves of both species, seven DEGs were identified, involving genes such as *SWEET17*, *DIR1*, *endochitinase EP3*, and *CCCH* zinc-finger protein 20 (Figure 7B). Detailed information regarding the expression levels of these genes is presented in Appendix A.

## 4. Discussion

Research on *S. glauca* and *S. salsa* has been primarily centered on their nutritional value, industrial applications, and ecological implications. Limited research has been conducted on the differences in saline–alkali tolerance mechanisms between these two species. Most studies thus far have only involved one species, with few comparative analyses between them. This study employed RNA-seq technology to analyze *S. glauca* and *S. salsa*, investigating their response mechanisms to saline–alkali stress at the transcriptional level. A total of 99,868 unigenes were obtained, with an average length of 993.35 bp. This study yielded more unigenes and richer data, indicating a higher sequencing quality and better assembly performance than previous studies on *S. glauca* or *S. salsa* [47,48]. Among these, 52,527 unigenes were successfully annotated, accounting for 52.60% of the total, with the highest number annotated to the NR database. This result also surpasses the percentage of NR annotation data in previous studies [49]. However, 47.40% of the unigenes in this study remained unannotated, suggesting that numerous novel transcripts remain to be further explored and identified.

The mechanisms underlying the differences in tolerance to saline–alkali between *S. glauca* and *S. salsa* remain unclear. We focused on studying specifically upregulated genes in the roots and leaves of *S. salsa* to elucidate these differences. There were 1189 and 1864 DEGs specifically upregulated in the roots and leaves of *S. salsa*, respectively. These genes may be key for the high saline–alkali tolerance of *S. salsa* than that of *S. glauca*. We conducted GO and KEGG analyses on these specifically upregulated genes to further investigate the discrepancies in tolerance mechanisms. The GO functional classification analysis indicated that the six categories with the highest enrichment of these genes were “metabolic process”, “cellular process”, “cell”, “cell part”, “binding”, and “catalytic activity”. This finding aligns with the study by Guo et al. [50]. The high enrichment of these genes in catalytic activity and binding suggests that plant molecular functions could be a primary factor contributing to the observed differences in plant salt tolerance [51].

According to the KEGG enrichment analysis, among all metabolic pathways, the DEGs involved in environmental information processing, genetic information processing, and metabolism constitute a significant proportion. The specifically upregulated genes in the roots and leaves of *S. salsa* are primarily enriched in key biological pathways, including “plant hormone signal transduction”, “starch and sucrose metabolism”, “carbon metabolism”, “biosynthesis of amino acids”, “oxidative phosphorylation”, “phenylpropanoid biosynthesis”, “amino sugar and nucleotide sugar metabolism”, and “MAPK signaling pathway-plant”. Multiple studies have demonstrated that pathways such as plant hormones [52], starch and sucrose biosynthesis [53], amino acid biosynthesis [54], MAPK signaling pathway [55], and phenylpropanoid biosynthesis [56] are crucial for plants in coping with adversity. Enriching these specifically upregulated genes in *S. salsa* roots and leaves suggests that the plant may enhance its ability to adapt to saline–alkali stress by strengthening these pathways.

“Carbon metabolism” and “starch and sucrose metabolism” play crucial roles in the energy metabolism of plants. Plants accumulate carbohydrates when external stresses exceed the impact of photosynthesis on plants, and their photosynthetic processes are also restricted by adversity stresses, such as drought and saline–alkali [57,58]. Plant sugars are key in stress perception and signal transduction, acting as regulatory hubs for gene expression in response to stress, ensuring proper osmotic adjustment, ROS scavenging, and maintaining cellular energy status through carbon allocation [59]. Liang et al. [60] utilized weighted gene co-expression network analysis (WGCNA), GO, and KEGG to analyze DEGs. This result suggests that ScDREB10 may influence starch and sucrose metabolism pathways to regulate plant tolerance to adversity stresses. Wang et al. [61] highlighted the significance of chloroplast starch and soluble sugars in the salt tolerance of salt-tolerant plants through proteomics. In this study, the specifically upregulated genes in the roots and leaves of *S. salsa* were enriched in “carbon metabolism” and “starch and sucrose metabolism”. This finding aligns with the results reported by Chen et al. [62]. It is speculated that these genes may be induced and expressed following saline–alkali stress, enhancing the resistance of *S. salsa* to such stresses.

Plant hormones, including auxin, ethylene, gibberellin (GA), ABA, and JA, play pivotal roles in plant stress responses. These hormones assist plants in adapting to a variety of stressful conditions by modulating growth, development, and metabolic processes [63]. Drought and high-salinity stresses increase ABA levels and the expression of stress-related genes, facilitating plant adaptation to challenging environments. Therefore, it is a key hormone for plant stress resistance [64]. Plants synthesize ABA via the carotenoid pathway, which begins with the cleavage of β-carotene, a C40 precursor, and through a series of transformations, xanthoxin, a C15 ABA precursor, is produced and subsequently converted into ABA by enzymes, such as *ABA2* and *AAO3* [63]. This study found four key genes involved in ABA biosynthesis that were specifically upregulated in the roots of *S. glauca* and *S. salsa*, with high expression levels in *S. salsa*. This suggests that the specific expression of these genes is linked to salt and alkaline tolerance in these plants and that *S. salsa* may enhance its tolerance by increasing the expression of ABA biosynthesis-related genes. The ABA signal transduction process primarily entails the recognition of ABA by receptors on the cell membrane, leading to the formation of complexes with *PP2C* and the activation of *SnRK2*. *SnRK2* phosphorylates downstream substrates, such as transcription factors and ion channels, to regulate diverse physiological responses in plants [65,66,67]. Several studies have reported that transcription factors, including *MYB* [68], *AP2/ERF* [69], *WRKY* [70], and *bHLH* [71], play crucial roles in plant salt stress resistance by modulating downstream gene expression. This study identified 11 downstream transcription factors regulated by *ABFs*, including three *ERF*, three *WRKY*, and five *bHLH* transcription factors. These transcription factors were upregulated in the roots of both *S. glauca* and *S. salsa*, with high expression levels in *S. salsa*. The ABA signal transduction pathway may regulate saline–alkali tolerance in these plants by modulating relevant transcription factors. JA plays a significant role in plant tolerance to saline–alkali stresses. Li et al. [72] identified *IbNAC087*, a core gene responding to salt and drought stresses, which activates the expression of JA synthesis-related genes *IbLOX* and *IbAOS*, enhancing the tolerance of sweet potatoes to salt and drought. In this study, we identified seven upregulated genes related to JA biosynthesis. This indicates that *S. salsa* may enhance its salt and alkaline tolerance by increasing the expression of JA biosynthesis-related genes.

To further elucidate the mechanistic disparities in defense mechanisms against saline–alkali stress between *S. salsa* and *S. glauca*, our study identified 24 DEGs in both roots and leaves that potentially respond to saline–alkali stress. Notably, previous research has established the involvement of *GPAT* [73], *TSJT1* [74], *HSFs* [75], *PEBP* [76], *SKOR* [77], *SWEET17* [78], *DIR1* [79], *TLPs* [80], and *CCCH zinc-finger proteins* [81] among these DEGs in plant stress responses to salt tolerance and drought resistance. In the present study, these DEGs were upregulated under saline–alkali stress, with *S. salsa* exhibiting higher expression levels than *S. glauca*. These findings suggest a strong correlation between these DEGs and the saline–alkali tolerance capabilities of both species, with the differential expression levels potentially accounting for their varied resilience to saline–alkali stress.

In summary, *S. salsa* may enhance its saline–alkali tolerance by increasing the biosynthesis of plant hormones, such as ABA and JA. This process strengthens the regulatory capacity of stress-resistance-related transcription factors, such as *WRKY*, *bHLH*, and *ERF,* and simultaneously elevates the expression levels of certain stress-resistance-related genes (Figure 8).

## 5. Conclusions

In summary, this study explored the transcriptome changes in leaves and roots in *S. glauca* and *S. salsa* using RNA-seq, obtaining transcriptome data of 155.84 Gb Clean Data. A total of 99,868 unigenes were obtained after assembly. Functional annotation of unigenes was performed, including comparison with NR, Swiss-Prot, KEGG, COG, KOG, GO, and Pfam databases, resulting in 52,527 annotation results for unigenes. Through GO and KEGG functional enrichment analysis, saline–alkali tolerant pathways and DEGs in the leaves and roots of *S. salsa* were identified. Key genes involved in ABA biosynthesis, signal transduction, and JA biosynthesis in *S. salsa* were screened, and 24 stress-resistant DEGs upregulated in both were further identified, which helped to understand the differences in salt-tolerant mechanisms between *S. glauca* and *S. salsa*. This study lays the research foundation for discovering the differences in genes and molecular mechanisms related to saline–alkali tolerance between *S. glauca* and *S. salsa*.

## Figures and Tables

**Figure 1 genes-15-01628-f001:**
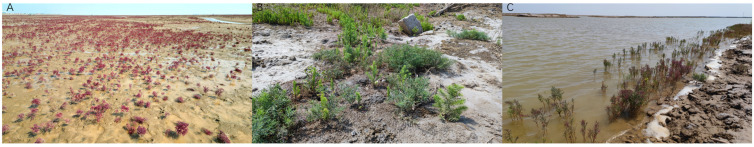
*S. salsa* and *S. glauca* growing in saline–alkali areas. (**A**) Red beach; (**B**) sampling location (Liujia Village, Dongji Town, Kenli District, Dongying City, Shandong Province, China); (**C**) *S. salsa* in flooding.

**Figure 2 genes-15-01628-f002:**
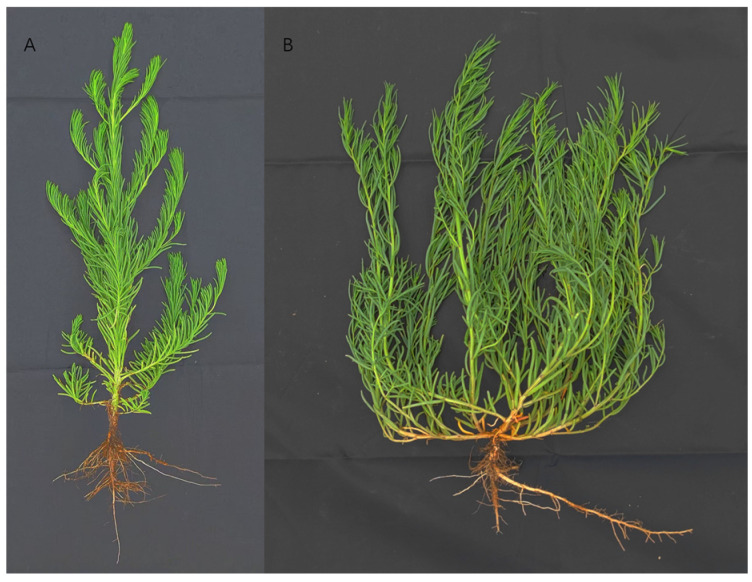
Morphological appearance of (**A**) *S. glauca*, and (**B**) *S. salsa*.

**Figure 3 genes-15-01628-f003:**
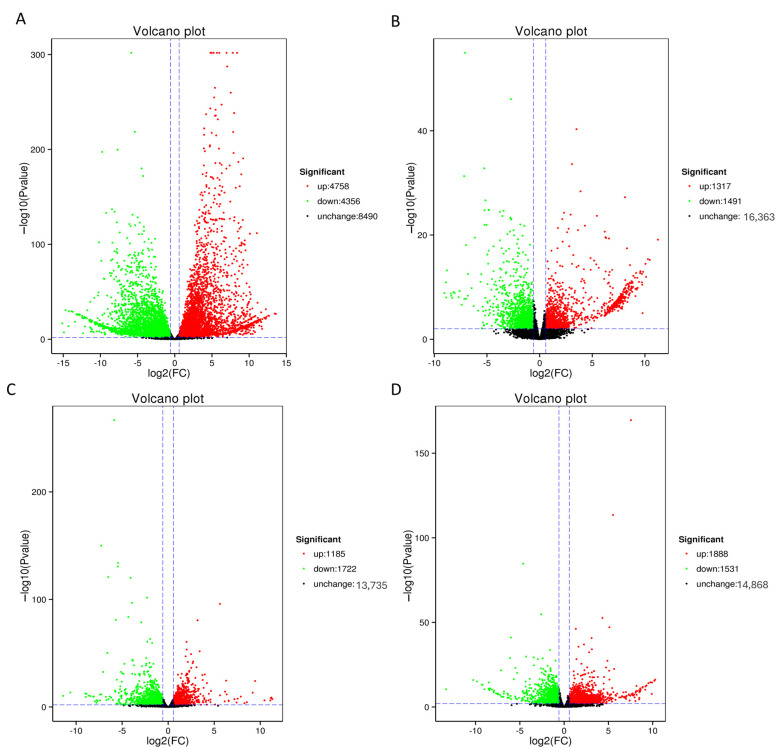
Differentially expressed genes (DEGs) in *S. glauca* and *S. salsa*. (**A**) DEGs in the roots of *S. glauca*, (**B**) DEGs in the roots of *S. salsa*, (**C**) DEGs in the leaves of *S. glauca*, and (**D**) DEGs in the leaves of *S. salsa*.

**Figure 4 genes-15-01628-f004:**
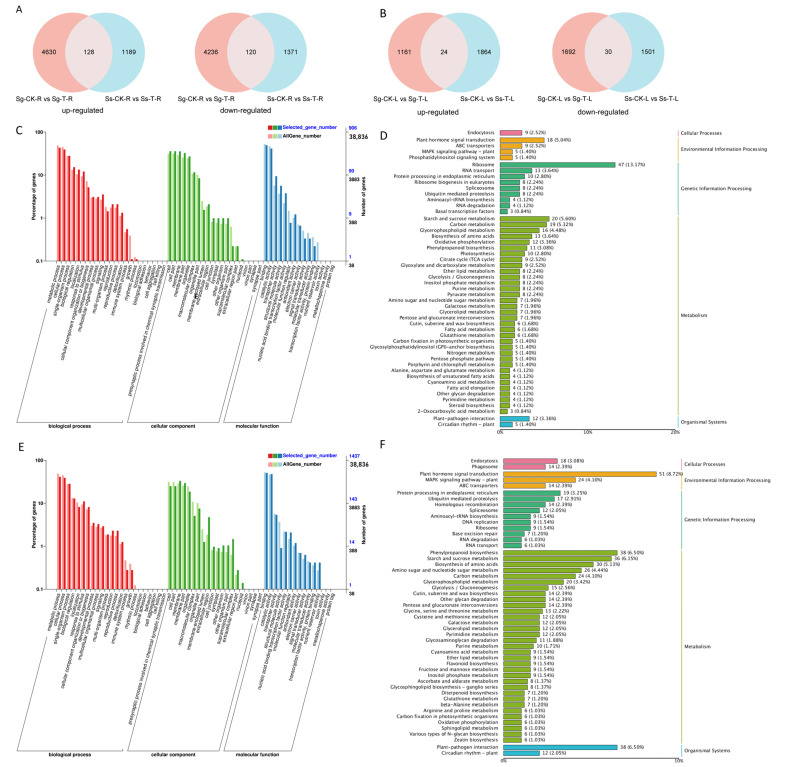
Analysis of differentially expressed genes (DEGs) in the roots and leaves of *S. glauca* and *S. salsa*. (**A**) DEGs in roots between *S. glauca* and *S. salsa*, (**B**) DEGs in leaves between *S. glauca* and *S. salsa*, (**C**) GO functional classification of specifically upregulated genes in *S. salsa* roots, (**D**) KEGG functional enrichment classification of specifically upregulated genes in *S. salsa* roots, (**E**) GO functional classification of specifically upregulated genes in *S. salsa* leaves, and (**F**) KEGG functional enrichment classification of specifically upregulated genes in *S. salsa* leaves.

**Figure 5 genes-15-01628-f005:**
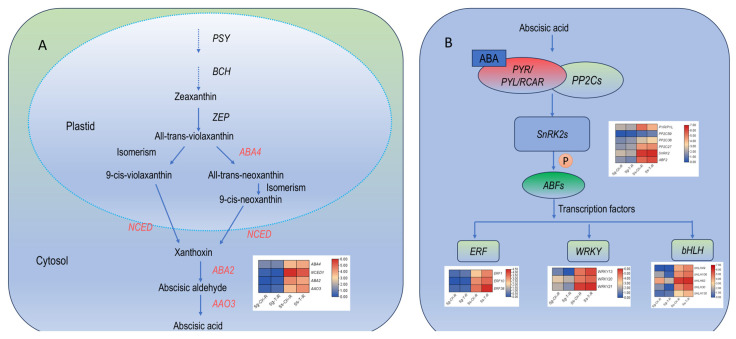
ABA biosynthesis and signal transduction in roots of *S. glauca* and *S. salsa*. (**A**) ABA biosynthesis pathway, and (**B**) ABA signal transduction.

**Figure 6 genes-15-01628-f006:**
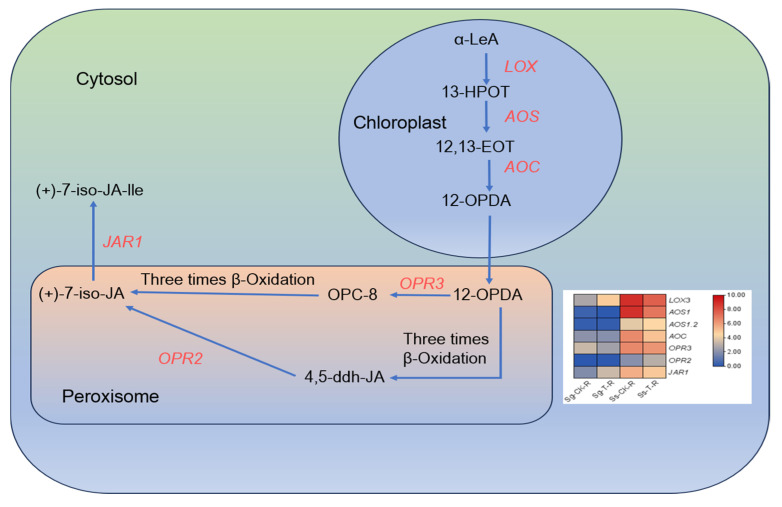
JA biosynthesis in roots of *S. glauca* and *S. salsa*.

**Figure 7 genes-15-01628-f007:**
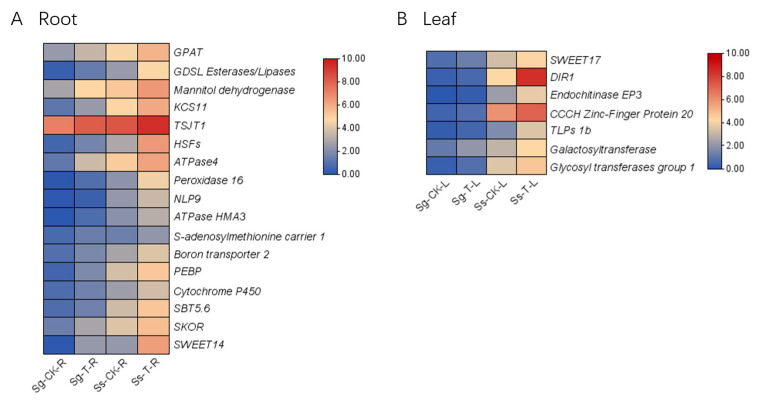
Candidate genes for saline–alkali tolerance in *S. glauca* and *S. salsa*. (**A**) Candidate genes related to saline–alkali tolerance in roots. (**B**) Candidate genes related to saline–alkali tolerance in leaves.

**Figure 8 genes-15-01628-f008:**
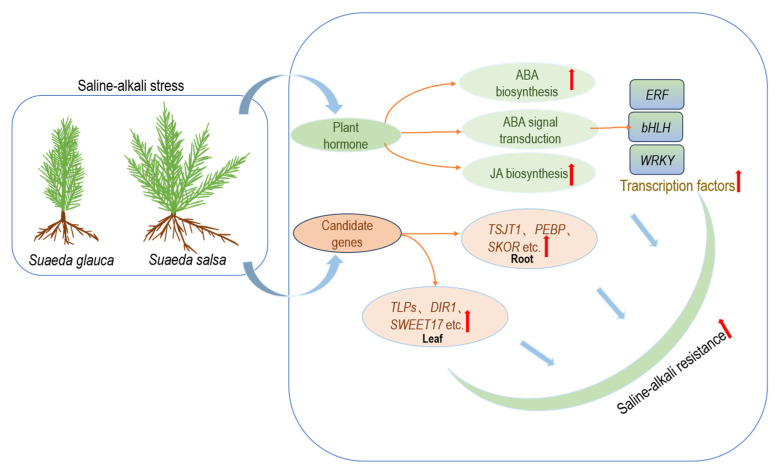
Schematic of saline–alkali tolerance mechanisms in *S. glauca* and *S. salsa*. The red arrow indicates that the relative expression level of the corresponding gene is up-regulated.

**Table 1 genes-15-01628-t001:** Assembly result statistics.

Length Range (bp)	All Unigenes	Sg	Ss
200–300	32,346 (32.39%)	23,823 (37.53%)	31,004 (39.05%)
300–500	25,655 (25.69%)	14,152 (22.30%)	17,542 (22.09%)
500–1000	13,275 (13.29%)	10,349 (16.30%)	12,685 (15.98%)
1000–2000	12,328 (12.34%)	7032 (11.08%)	9121 (11.49%)
2000+	16,264 (16.29%)	8117 (12.79%)	9045 (11.39%)
Total Number	99,868	63,473	79,397
Total Length	99,204,098	56,364,907	64,912,814
N50 Length	2205	1964	1716
Mean Length	993.35	888.01	817.57

**Table 2 genes-15-01628-t002:** Statistics of annotated gene information from various databases.

Annotated Database	Annotated Number	300 bp ≤ Length	Length ≥ 1000 bp
COG_Annotation	14,193	2688	9220
GO_Annotation	38,836	9731	21,949
KEGG_Annotation	31,422	7187	18,911
KOG_Annotation	26,358	5957	15,531
Pfam_Annotation	35,881	8355	21,966
Swissprot_Annotation	29,004	6160	19,207
TrEMBL_Annotation	45,556	11,737	26,045
eggNOG_Annotation	37,071	9066	22,573
nr_Annotation	50,760	13,778	26,301
All_Annotated	52,527	14,345	26,426

## Data Availability

Data are contained within the article and Appendix A.

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
