# Peer review of "Comparative Transcriptome Analysis Reveals Mechanisms of Differential Salinity Tolerance Between Suaeda glauca and Suaeda salsa"

_genes, 2024, doi:10.3390/genes15121628_

Round 1

Reviewer 1 Report

Comments and Suggestions for Authors

Dear authors

In the present work, the authors have approached the comparative transcriptome analysis to study the molecular mechanism of Suaeda glauca  and  Suaeda salsa rice under salinity conditions (saline−alkali soil). The study identified 12,021 and 6,227 differentially expressed genes (DEGs) from the S. glauca and S. salsa under aline-alkali stress, respectively. Among these DEG, the authors identified key genes involved in stress-related pathways such as ABA and JA biosynthesis that significantly enhance their adaptation to saline-alkali stress.

The data was obtained using appropriate analytical methods and the obtained results are convincing. I have some suggestions and a few minor remarks.

Comments:

1. Please double-check and revise the notation/legend for Fig 2. I think there is a mislabeling here. According to the description in the text “the branching ability of S. salsa is stronger than that of S. glauca, while its upright ability is weaker than that of S. glauca.” (Line 208-209), the notation for Fig2A should be “Suaeda glauca” and Fig2B should be “Suaeda salsa”.

2. Since Fig 1 has multiple panels listed as A, B, and C. However, there is a lack of citation of Figures (1A and 1B) in the main text, only Fig1C was cited.

3. Re-interpret the results of Table 1. The authors stated in the main text that “Among these, unigenes longer than 500 bp accounted for 38.98% of the total in the library " (Lines 240-241). However, I see “13.29%” in Table 1. Please double-check and revise this.

Minor remarks:

- Revise the sentence (Line 31-32): “Notably, 1,189 and 1,864 were specifically upregulated DEGs in the roots and leaves of S. salsa under saline−alkali conditions.” To “Notably, 1,189 and 1,864 were specifically upregulated DEGs in the roots and leaves of S. salsa under saline−alkali conditions, respectively

- When the first place in the text is presented in abbreviations, their full form of abbreviations should be presented, afterward abbreviations can be presented instead of full form. For instance, abscisic acid (ABA) and jasmonic acid (JA) (Line 330) should be presented early in the Abstract section (Line 35). 

- Add “S. glauca” to legend of Fig 4 and Fig 8 since these Figures including the S. glauca.

For instance, “Figure 4. Analysis of differentially expressed genes (DEGs) in the roots and leaves of S. salsa.” revised to “Figure 4. Analysis of differentially expressed genes (DEGs) in the roots and leaves of S. salsa and S. glauca

- Italicize the gene names: SOS1, SOS2, and SOS3 (Line 78) and plant: Chenopodiaceae (Line 97).

- Overall, the manuscript is readable and easy to understand except for a few sentences, which are repetitive, complicated, and lack clarity. I recommend a thorough review for grammar errors, repeated word, and sentence structure to enhance the readability, clarity and precision of this paper.

For instance, in the section 2.4, there are many repetition of “removed”; the Fig 2’s legend; and section 3.8 (Line 365-366) “…between S. glauca and S. salsa. Between the roots of S. glauca and S. salsa,…”

The sentence (Line 159-160) “After passing the sample inspection, Beijing Baimaike Bioinformatics Technology Co., Ltd. constructed and controlled the cDNA library.” It lacks clarity, thus it should be represented.

I have marked the above comments on the manuscript. Please use it for easy tracking and revising.

Best regards, 

Author Response

We are extremely grateful for the time and effort you invested in reviewing our manuscript. Your professional and astute suggestions have been highly valuable. We have meticulously revised the manuscript following your advice, and we are confident that these revisions have substantially enhanced the quality and clarity of our work. Below are our point-to-point responses to reviewers’comments.

Comment1: Please double-check and revise the notation/legend for Fig 2. I think there is a mislabeling here. According to the description in the text “the branching ability of S. salsa is stronger than that of S. glauca, while its upright ability is weaker than that of S. glauca.” (Line 208-209), the notation for Fig2A should be “Suaeda glauca ” and Fig2B should be “Suaeda salsa”.

Response1: We thank the reviewer for pointing this out. After checking, the legend for Figure 2 has been rectified to align with the text, with ‘Suaeda glauca’ now correctly assigned to Fig 2A and ‘Suaeda salsa’ to Fig 2B.

Comment2: Since Fig 1 has multiple panels listed as A, B, and C. However, there is a lack of citation of Figures (1A and 1B) in the main text, only Fig1C was cited.

Response2: We thank the reviewer for the suggestion. In our revised manuscript, We have made reasonable citations for Figure 1A and Figure 1B at lines 220 and 222 in the article.

Comment3: Re-interpret the results of Table 1. The authors stated in the main text that “Among these, unigenes longer than 500 bp accounted for 38.98% of the total in the library” (Lines 240-241). However, I see “13.29%” in Table 1. Please double-check and revise this.

Response: We thank the reviewer for pointing this out. Upon re-examination, 38.89% in the manuscript was the proportion of unigenes with a length ranging from 300 to 1000 bp. We have corrected it to "unigenes longer than 500 bp accounted for 13.29% of the total in the library" in the revised manuscript.

Comment4: Revise the sentence (Line 31-32): “Notably, 1,189 and 1,864 were specifically upregulated DEGs in the roots and leaves of S. salsa under saline−alkali conditions.” To “Notably, 1,189 and 1,864 were specifically upregulated DEGs in the roots and leaves of S. salsa under saline−alkali conditions, respectively”.

Response: We thank the reviewer for pointing this out. As suggested, we have made the corresponding revisions in our revised manuscript.

Comment5: Line 19: When the first place in the text is presented in abbreviations, their full form of abbreviations should be presented, afterward abbreviations can be presented instead of full form. For instance, abscisic acid (ABA) and jasmonic acid (JA) (Line 330) should be presented early in the Abstract section (Line 35).

Response5: We thank the reviewer for the suggestion. As suggested, we corrected abscisic acid (ABA), jasmonic acid (JA), and other similar terms in our revised manuscript.

Comment6: Add “S. glauca” to legend of Fig 4 and Fig 8 since these Figures including the S. glauca. For instance, “Figure 4. Analysis of differentially expressed genes (DEGs) in the roots and leaves of S. salsa.” revised to “Figure 4. Analysis of differentially expressed genes (DEGs) in the roots and leaves of S. salsa and S. glauca”.

Response6: We thank the reviewer for the suggestion. As suggested, we added "S. glauca" to the legends of Figure 4 and Figure 8.

Comment7: Italicize the gene names: SOS1, SOS2, and SOS3 (Line 78) and plant: Chenopodiaceae (Line 97).

Response7: Corrected.

Comment8: Overall, the manuscript is readable and easy to understand except for a few sentences, which are repetitive, complicated, and lack clarity. I recommend a thorough review for grammar errors, repeated word, and sentence structure to enhance the readability, clarity and precision of this paper. For instance, in the section 2.4, there are many repetition of “removed”; the Fig 2’s legend; and section 3.8 (Line 365-366) “…between S. glauca and S. salsa. Between the roots of S. glauca and S. salsa,…”. The sentence (Line 159-160) “After passing the sample inspection, Beijing Baimaike Bioinformatics Technology Co., Ltd. constructed and controlled the cDNA library.” It lacks clarity, thus it should be represented.

Response8: We thank the reviewer for the suggestion. As suggested, we have carefully examined and corrected grammatical mistakes, eliminated redundant words, and improved sentence structures throughout the text. For instance, the sentence “After passing the sample inspection, Beijing Baimaike Bioinformatics Technology Co., Ltd. constructed and controlled the cDNA library” has been revised to “After sample inspection, library construction and quality control were performed by BioMarker Technologies Company (Beijing, China).”

Reviewer 2 Report

Comments and Suggestions for Authors

The paper performs a very interesting comparison among two Sueade species to get insights into the differences among transcriptomic responses. The study is sound but requires some improvements before being suitable for publication.

The authors have focused a lot on their results, but the introduction requires further information to help understand the results. My main criticism in this aspect is: What is known about the mechanism of these two species under salt stress? Do they accumulate sodium in leaves or extrude at the root level? Do they synthesize any osmolite to prevent water loss? Is the accumulation in the vacuole? Do they increase potassium transport to counteract sodium uptake? Please make a brief description of what is known.

In the results section and the conclusion, the focus is on ABA and JA, but what happens with the other hormones? For instance, a recent paper has pinpointed that in some plants the key factor for tolerance may be cytokinine signalling. 

https://onlinelibrary.wiley.com/doi/10.1111/jipb.13755?s=08

Please comment about how the cytokines are in this experiment.

Another issue is that the number of differentially expressed genes (DEGs) reported for S. glauca and S. salsa varies across different sections. At some point it is mentioned that S. glauca had 12,021 DEGs and S. salsa had 6,227 DEGs under salt stress. However, in another section, it is stated that S. salsa had 1,189 DEGs in roots and 1,864 in leaves specifically upregulated under saline−alkali conditions. Please clarify.

Other points:

There is a large number of non-annontated genes. May this vary the interpretation of the results? Are there non-annotated genes among the DEGs? Please comment about this. 

Figure 3 and Figure 4: Nothing can be read. Try to enlarge the figures and increase the font size. 

Author Response

We are grateful for the valuable suggestions provided by the reviewer. We have carefully and thoroughly revised the manuscript to address the reviewer’s comments and suggestions. Below are our point-to-point responses to reviewers’comments.

Comment1: The authors have focused a lot on their results, but the introduction requires further information to help understand the results. My main criticism in this aspect is: What is known about the mechanism of these two species under salt stress? Do they accumulate sodium in leaves or extrude at the root level? Do they synthesize any osmolite to prevent water loss? Is the accumulation in the vacuole? Do they increase potassium transport to counteract sodium uptake? Please make a brief description of what is known.

Response1: We thank the questions raised by the reviewer. In the revised manuscript, we incorporated the research findings regarding the mechanisms of S. glauca and S. salsa under salt stress in the introduction section (lines 110 - 133). Under saline-alkali stress conditions, both S. glauca and S. salsa can alleviate the impact of salt stress through means such as leaf succulence, ion compartmentalization, osmotic adjustment and the antioxidant system [29-31]. Salt treatment can significantly promote the absorption of Na⁺ in the aboveground parts of S. salsa and S. glauca, and significantly reduce the absorption of K⁺ in the aboveground parts. Moreover, the con-tent of Na⁺ in the aboveground parts of S. salsa is significantly higher than that of S. glauca, while the content of K⁺ is significantly lower than that of S. glauca, which demonstrates the differences in their salt tolerance mechanisms [32]. Different salt concentrations have different effects on the growth and leaf succulence of S. salsa seed-lings, among which Na⁺ and Cl⁻ are the two main factors promoting the succulence of S. salsa leaves [33]. Compared with S. glauca, S. salsa accumulates more total nitrogen (TN), total organic nitrogen carbon (TOC), sodium (Na) and so on in its leaves. During the process of adapting to salinity, the central metabolism of S. glauca is nitrogen metabolism, while that of S. salsa is characterized by organic acid metabolism [34]. During the growth period of S. salsa, a large amount of Na⁺ will accumulate in its leaves. S. salsa can regulate the Na⁺ balance in its body and maintain the accumulation level of Na⁺ through SsSOS1 and SsNHX1 [35]. The localization analysis of Na⁺ in S. salsa after salt stress revealed that it isolates most of the Na⁺ absorbed from the environment in the vacuoles to maintain the normal ionic balance within cells [36]. Under high Na⁺ or high K⁺ conditions, S. salsa has a specific absorption system for K⁺. While absorbing a large amount of Na⁺, it can also effectively absorb K⁺. This characteristic of K⁺ absorption by S. salsa is one of the important factors for its normal growth under high salinity conditions [37].

Comment2: In the results section and the conclusion, the focus is on ABA and JA, but what happens with the other hormones? For instance, a recent paper has pinpointed that in some plants the key factor for tolerance may be cytokinine signalling.

https://onlinelibrary.wiley.com/doi/10.1111/jipb.13755?s=08

Please comment about how the cytokines are in this experiment

Response2: We thank the reviewer for asking the suggestion. To verify whether there were differences in cytokinins between S. glauca and S. salsa under saline-alkali stress, we visualized the relative expression levels of the key enzymes for the de novo synthesis of cytokinins, including isopentenyl transferases (IPTs), LONELY GUYs (LOGs) and cytokinin oxidases (CKXs). Under saline-alkali stress, different from the situation where the relative expression levels of abscisic acid (ABA) and jasmonic acid (JA) were higher in S. salsa than in S. glauca, cytokinins did not exhibit the same pattern as ABA and JA between S. glauca and S. salsa, and there were no significant differences in their expression levels between S. glauca and S. salsa. This suggests that cytokinins may not be the principal plant hormones playing a major role in the saline-alkali tolerance of S. glauca and S. salsa, while ABA and JA play crucial roles in S. glauca and S. salsa.The visualization results of the relative expression levels of relevant genes have been attached for your reference.

Comment3: Another issue is that the number of differentially expressed genes (DEGs) reported for S. glauca and S. salsa varies across different sections. At some point it is mentioned that S. glauca had 12,021 DEGs and S. salsa had 6,227 DEGs under salt stress. However, in another section, it is stated that S. salsa had 1,189 DEGs in roots and 1,864 in leaves specifically upregulated under saline−alkali conditions. Please clarify.

Response3: We thank the reviewer for this comment. Under saline-alkali stress, the numbers of differentially expressed genes in the roots and leaves of S. glauca were 9114 and 2907 respectively, and those in the roots and leaves of S. salsa were 2808 and 3419 respectively. That is to say, 12,021 and 6,227 DEGs were the total numbers of differentially expressed genes in the roots and leaves of S. glauca and S. salsa respectively. And 1,189 and 1,864 DEGs were the differentially expressed genes that were only up-regulated in the roots and leaves of S. salsa under saline-alkali stress. These genes may be the key genes contributing to the differences in saline-alkali tolerance between S. glauca and S. salsa.

Comment4: There is a large number of non-annontated genes. May this vary the interpretation of the results? Are there non-annotated genes among the DEGs? Please comment about this.

Response4: We thank the reviewer for asking this question. Due to the fact that our transcriptome data was sequenced relatively early and lacked the reference genome of this species, many genes were not annotated. We have made every effort to annotate the genes using multiple databases. Among numerous differentially expressed genes (DEGs), there are also a small number of unannotated genes, which is similar to the results of other researchers' transcriptome studies on S. glauca. Under saline-alkali stress, the differential genes were enriched in pathways related to abiotic stress, such as "Plant hormone signal transduction", "Oxidative phosphorylation", "Phenylpropanoid biosynthesis" and "MAPK signaling pathway - plant". This result is also similar to the enrichment results of other species under salt stress. Therefore, we believe that our results are still of great significance for revealing the saline-alkali tolerance mechanisms between S. glauca and S. salsa.

Comment5: Figure 3 and Figure 4: Nothing can be read. Try to enlarge the figures and increase the font size.

Response5: We thank the reviewer for the suggestion. We have already adjusted the sizes of the pictures and texts of Figure 3 and Figure 4 in the revised manuscript.

References

  1. Li, H.; Wang, H.; Wen, W.; Yang, G. The antioxidant system in Suaeda salsa under salt stress. Plant Signal. Behav. 2020, 15, 1771939.
  2. Ping-Hua, L.; Zeng-Lan, W.; Hui, Z.; Bao-Shan, W. Cloning and Expression Analysis of the B Subunit of V-H + -ATPase in Leaves of Halophyte Suaeda salsa Under Salt Stress. Acta Botanica Sinica. 2004, 46.
  3. Yang, C.; Shi, D.; Wang, D. Comparative effects of salt and alkali stresses on growth, osmotic adjustment and ionic balance of an alkali-resistant halophyte Suaeda glauca (Bge.). Plant Growth Regul. 2008, 56, 179-190.
  4. Wu, Y; Feng, X; Zang, R; Li J; Liu, X. Comparative study on the growth and Cd uptake of Suaeda salsa and Suaeda glauca under the stress of salt and Cd and their interaction. Chin. J. Eco-Agric. 2022, 30, 1186-1193.
  5. Qi, C.; Chen, M.; Song, J.; Wang, B. Increase in aquaporin activity is involved in leaf succulence of the euhalophyte Suaeda salsa, under salinity. Plant Sci. 2009, 176, 200-205.
  6. Song, X.; Yang, N.; Su, Y.; Lu, X.; Liu, J.; Liu, Y.; Zhang, Z.; Tang, Z. Suaeda glauca and Suaeda salsa Employ Different Adaptive Strategies to Cope with Saline–Alkali Environments. Agronomy. 2022, 12, 2496.
  7. Wang, W.; Liu, Y.; Duan, H.; Yin, X.; Cui, Y.; Chai, W.; Song, X.; Flowers, T.J.; Wang, S. SsHKT1;1 is coordinated with SsSOS1 and SsNHX1 to regulate Na+ homeostasis in Suaeda salsa under saline conditions. Plant Soil. 2020, 449, 117-131.
  8. Qiu, N.; Chen, M.; Guo, J.; Bao, H.; Ma, X.; Wang, B. Coordinate up-regulation of V-H+-ATPase and vacuolar Na+/H+ antiporter as a response to NaCl treatment in a C3 halophyte Suaeda salsa. Plant Sci. 2007, 172, 1218-1225.
  9. Mori, S.; Suzuki, K.; Oda, R.; Higuchi, K.; Maeda, Y.; Yoshiba, M.; Tadano, T. Characteristics of Na+ and K+ absorption in Suaeda salsa (L.) Pall. Soil Sci. Plant Nutr. 2011, 57, 377-386.
